# Prediction of Deflection Due to Multistage Loading of a Corrugated Package

**Jong-Min Park** [1], **Tae-Yun Park** [2,*] and **Hyun-Mo Jung** [3,*]

1   Department of Bio-Industrial Machinery Engineering, Pusan National University, Miryang 50363, Republic of Korea
2   H & A NPI VPD Project, LG Electronics, Changwon 51533, Republic of Korea
3   Department of Logistic Packaging, Kyungbuk Science College, Chilgok 39913, Republic of Korea
*   Correspondence: taeyun.park@lge.com (T.-Y.P.); hmjung@kbsc.ac.kr (H.-M.J.)

**Abstract:** With the expansion of overseas markets, transportation distance, storage periods in warehouses, and traffic volume of goods are increasing. In this distribution environment, the safety problem due to the decrease in the strength of the multistacked corrugated package is becoming very important. This study aims to develop a CAE prediction technology for the height change of multistacked corrugated packages, and for this study, the FEA simulation method for existing corrugated packages has been investigated and supplemented. The four-point bending FE model and the box compression test FE model were also constructed based on the simplified and homogenized composite model for the target corrugated fiberboard (double wall of EB-flute) itself. Four-point bending, box compression, and creep tests were performed to obtain the material constant required for FEA simulation. Through the comparison of the F–D curve area between the test and the FEA simulation, parameter optimization that minimizes the area difference was performed using Hyper-Study. The FEA simulation and stacking test for the multistacked target corrugated package were performed simultaneously on four actual stacking scenarios with different package weights and package sizes. A comparison of height changes after 72 h of stacking for each of the four scenarios showed that the concordance between the test and FEA simulation was more than 80% in all cases. To further expand the scope of this application, it is necessary to secure additional reliability through continuous comparative monitoring using the test data and physical properties of various corrugated fiberboards.

**Keywords:** corrugated fiberboard; box creep; finite element analysis; corrugated package

## 1. Introduction

Corrugated packages are often crushed or collapse at lower levels when the shipped products are stored in a multistage loading state, resulting in financial losses. To prevent this, many industrial sites conduct package-stacking tests on actual products. Generally, package-stacking tests are based on transportation and storage; the temperature and humidity conditions corresponding to these distribution environments are considered as the test conditions, and the height reduction over time is considered as the basis for determining the success or failure of the test.

The McKee equation [1] has been traditionally used to predict the compression strength of corrugated packages; however, it only predicts the static strength, not the difference in the height over time. Owing to the nature of modern logistics, the transportation distance, storage period, and traffic volume of goods are increasing. Therefore, the problem of strength reduction and compromised safety because of multistacked corrugated packages has become critical. Therefore, it is highly essential to correlate the package compression strength with the stacking strength of corrugated packages while considering the actual distribution conditions.

Various studies have been conducted to accurately predict the mechanical behavior of corrugated packages and fiberboards; in addition, prediction methods using finite element analysis (FEA) have been suggested. A parametric study on the buckling strength of corrugated fiberboards accounts for a significant portion of the application of FEA [2]. Nordstrand [3] applied FEA to analyze the post-buckling strength of a corrugated fiberboard panel subjected to edge compressive loading, and Fadiji et al. [4] analyzed the critical buckling load and buckling shape of an edgewise crush resistance test (ECT) model of a corrugated fiberboard through FEA. In addition, some studies [5,6] have applied FEA to analyze the strength of various test specimens used in the international standard ECT of corrugated fiberboard. The FEA model has also been applied to analyze the bending stiffness [7–9], flat crush resistance [2], and dynamically advantageous board combination of corrugated fiberboards [10].

The application of the FEA simulation to the design optimization of corrugated packages through the analysis of various mechanical behaviors of corrugated fiberboards has contributed significantly to the literature, as it can cover areas beyond the limits of experimental approaches. However, the long time required for modeling and conducting the FEA of complex corrugated structures, as well as limitations in applying FEA material constants and convergence problems in FEA, has been pointed out by several researchers, and several efforts have been made to solve these problems. One solution is to represent complex corrugated structures as simplified models and determine their equivalent (effective) mechanical properties [2,11–13].

Corrugated fiberboard, a material used for fabricating packaging boxes, is a composite material manufactured by stacking liners, and the corrugating medium has anisotropic properties. Although the geometric properties of the corrugated fiberboard can be considered in detail when analyzing its properties, it is extremely difficult to model because of the structure of folds made while manufacturing the box, wherein the corrugated fiberboard is cut and folded through scoring and creasing. To overcome this, a method was proposed that assumed that the region of the wavy corrugating medium (flute) between the liner and the liner of the corrugated fiberboard was homogenized with a simple volume to define the corresponding material constants [14,15].

Jiménez-Caballero [14] applied the Coffin creep model to simulate the sagging of the box's bottom owing to the weight of the packaged goods using Abaqus S/W, and stated that the continuum shell element type was more suitable than the conventional shell element type for predicting the sagging over time. In addition, connector elements with joint and revolute properties were applied to describe the behavior of the box folds.

In this study, the results of the aforementioned prior studies were applied to a height reduction prediction FEA simulation via the multistage loading of a target corrugated package, and the power-law model was used instead of the Coffin model as the creep model. The Coffin and power-law models are shown in Equations (1) and (2), respectively [14,16].

$$\varepsilon^{cr} = \frac{\sigma_o}{E}\left[1 + A\left(1 - e^{-at^\alpha}\right) + B\,ln(bt + 1)\right] \tag{1}$$

where $\varepsilon^{cr}$ represents the creep strain over time; $\sigma_o$ is the initial stress; $E$ is the elastic modulus; $t$ is the time; and $A$, $B$, $a$, $b$, and $\alpha$ represent the material constants.

$$\varepsilon^{cr} = \frac{1}{m+1}A\sigma_o{}^n t^{m+1} \tag{2}$$

where $A$, $m$, and $n$ represent the material constants.

The purpose of this research was to validate the FEA simulation technique for predicting height reduction through the multistage loading of a target corrugated package. The FEA material constants required for the simulation were obtained through various tests, and the physical property parameters were calculated through an optimization simulation using Altair HyperStudy to minimize the errors in the test and FEA. The suitability of the method was verified by comparing the package-stacking test results with those of the FEA.

## 2. Finite Element Modeling of a Corrugated Package

As shown in Figure 1, a corrugated fiberboard was manufactured by bonding the wavy corrugating medium (flute) between the liners. In the production process, the direction of the flow of the equipment is the machine direction (MD), the direction orthogonal to the flat MD is the count machine direction (CD), and the direction perpendicular to the corrugated fiberboard's surface is the thickness direction (z-direction).

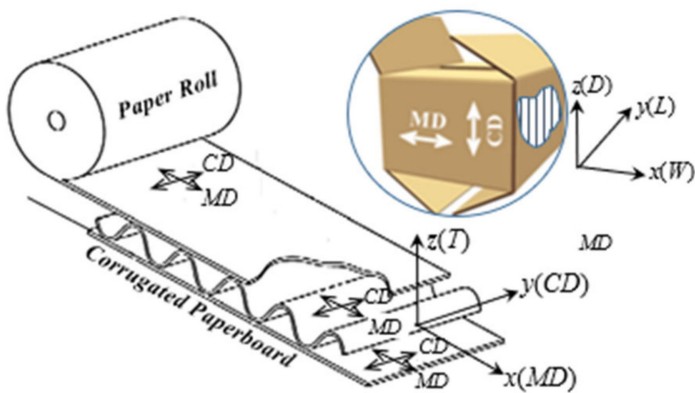

**Figure 1.** Definitions of the directions of the corrugated fiberboard and corrugated package [5,13]: CD, count machine direction; MD, machine direction.

As shown in Figure 1, the liner and corrugating media have orthotropic properties during the manufacturing process; however, the combined corrugated fiberboard also has orthotropic properties because of the unique characteristics of the flute. In addition, the tensile and compressive direction characteristics are different and exhibit nonlinear properties [13]. Therefore, in this study, we assumed that the corrugated package has a linear behavior. The corrugated fiberboard is classified into several types according to the number of combinations of liners and corrugating media, as well as the shape of the applied flute. In this study, the DW corrugated fiberboard of the EB-flute (EB/F) was targeted. The DW corrugated fiberboard of the EB/F consists of two flutes (E-flute (E/F) and B-flute (B/F)) and three liners (outer, middle, and inner liner), as shown on the left side of Figure 2. The flute height and flute pitch of E/F range from 1.4 to 1.6 mm and 3.33 to 3.19 mm, respectively, and the flute height and flute pitch of E/F range from 2.5 to 2.8 mm and 6.25 to 5.77 mm, respectively. The board combination of the target corrugated fiberboard was (SC300 surface coating+) KA180/K180(E/F)/S120/K180(B/F)/K180 (SC: outer BKP + inner Manila waste paper (BKP—bleached kraft pulp); KA: 30% outer liner containing UKP + 40% AOCC + 30% KOCC (UKP—unbleached kraft pulp; AOCC—American old corrugated container; KOCC—Korean old corrugated container, K180 and S120; 100% KOCC)). The measured total thickness of the corrugated fiberboard was approximately 5 mm. Abaqus laminar material constants were determined to define the anisotropic properties of the liner, assuming that the areas of the E/F corrugating medium, the intermediate liner, and the B/F corrugating medium between the outer and inner liners were combined for simplification and homogenization and were defined as the core. The core was also defined by the same Abaqus laminar material constants as those of the liner. As shown in Figure 2, the laminated structure of the liner and the corrugating medium was defined by the Abaqus composite model and consisted of an outer liner, core, and inner liner. In the simplified model of Figure 2, *t* is the ratio of the thickness occupied by the liner and the core with a total thickness of the corrugated fiberboard of 5 mm. To simplify the model, we considered the thicknesses of the outer and inner liners to be equal, with an average thickness of 0.2 mm.

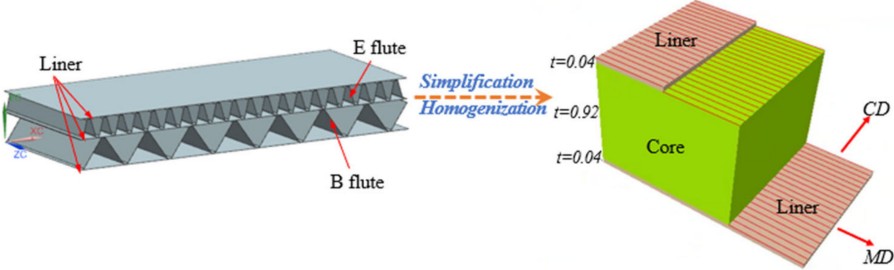

**Figure 2.** Simplified and homogenized EB/F corrugated fiberboard composite model.

The power-law creep model of Equation (2) was applied to predict the time-dependent deflection with respect to the compression load of the target corrugated package, wherein we assumed that the creep behaviors of the liner and core were the same. The box was modeled in 3D based on the drawings of the target corrugated package. As Jiménez-Caballero [14] mentioned, the continuum shell element was applied as the basic element type of the corrugated fiberboard because it is more suitable for predicting the changes over time than conventional shell elements and advantageous for handling contact between the packages. Because the compressive deformation of the corrugated package depends on the rigidity of the edge and corner of the package, detailed modeling of this component was conducted, as shown in Figure 3. It was modeled as a connector (CONN3D2) element to represent the rotation characteristics of the folded component, and the degrees of freedom of rotation (revolute) and spring characteristics were given [16]. To connect the connector and the continuum shell element, a conventional shell element was inserted in one row, and the two shell element types were bound through shell-to-solid coupling. Shell-to-solid coupling in Abaqus is a surface-based technique for coupling shell elements to solid elements. Shell-to-solid coupling assembles constraints that couple the displacement and rotation of each shell node to the average displacement and rotation of the solid surface in the vicinity of the shell node [16].

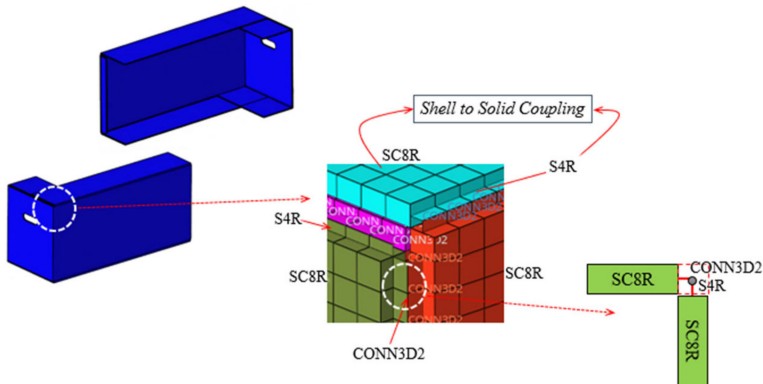

**Figure 3.** Target corrugated package and detailed modeling of the folds.

## 3. Material Property Testing

To define the material constants ($E_1$, $E_2$, $v_{12}$, $A$, $n$, and $m$) for simulating the prediction of deflection due to the multistage loading of the corrugated package, three material property tests were performed, as described in the following subsections.

### 3.1. Four-Point Bending Test

Specimens with the dimensions of $50 \times 500$ mm were manufactured similarly to the target corrugated package. The prepared specimens were pretreated in a chamber under severe conditions (40 °C at an rh of 90%) for more than 72 h; then, a four-point bending test was conducted at room temperature (Figure 4). The test followed the TAPPI T820 [17] specifications, and the test speed was 25.4 mm/min.

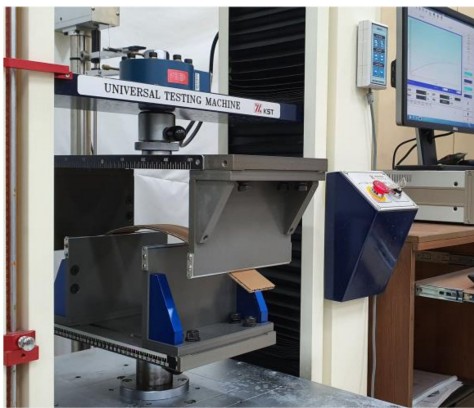

**Figure 4.** Four-point bending test apparatus for the corrugated fiberboard specimen.

The results of the four-point bending test are shown as the bending force (F)–deflection (δ) curve of the MD and CD in Figure 5. Here, δ is the deflection of the specimen at the central span, which is one-third of the crossed-head movement of the universal testing machine [10]. In this figure, when E/F > B/F, the E/F side becomes tensile and the B/F side becomes compressed, whereas when B/F > E/F, the B/F side becomes tensile and the E/F side becomes compressed.

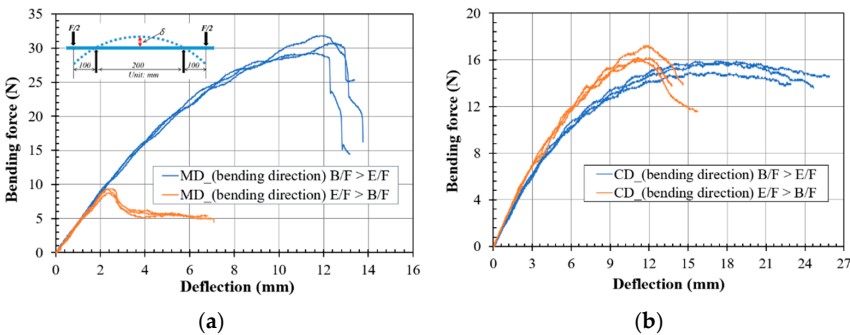

(**a**)  (**b**)

**Figure 5.** Bending force–deflection curves from the four-point bending test. (**a**) Machine direction (MD). (**b**) Count machine direction (CD).

### 3.2. Uniaxial Box Compression Test

The box compression test (BCT) was performed at room temperature (Figure 6) after pretreating the sample for 72 h or more under two conditions: (1) temperature of 40 °C at an rh of 50%, and (2) temperature of 40 °C at an rh of 90%. The test followed the ISO 12048 [18] specification and was conducted at a test speed of 10 mm/min for a box with a size of 730 × 293 × 336 mm based on the outer dimensions. The box sample used in the BCT was made with a target corrugated fiberboard and was a regular slotted container (RSC) with a size of L × W × D = 730 × 293 × 336 mm based on the outer dimensions.

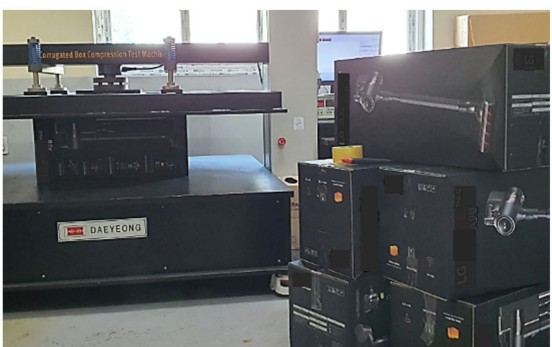

**Figure 6.** Uniaxial box compression testing apparatus.

The results of the BCT were expressed as a load–deflection curve, as shown in Figure 7.

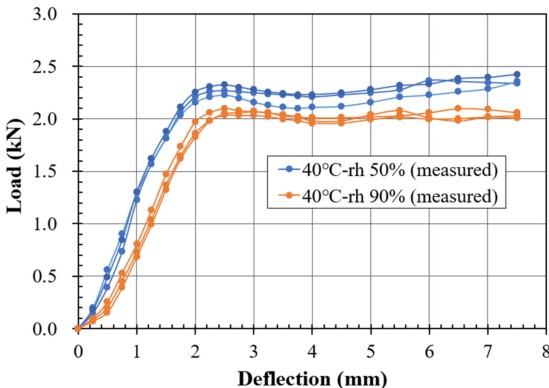

**Figure 7.** Load–deflection curves from the uniaxial box compression test.

### 3.3. Box Creep Test

As shown in Figure 8, the box creep test was conducted for 72 h with a compression load of 0.8338 kN (approximately 41% of the box compression strength under the same conditions (Figure 7)) at a temperature of 40 °C and an rh of 90%, considering the extreme distribution conditions of the target corrugated package. The sample used was the same as that used in the BCT, and the test followed the ASTM D7030 [19] specifications.

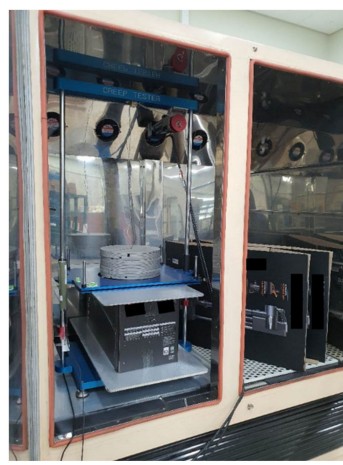

**Figure 8.** Box compression creep testing apparatus.

The results of the box creep test were expressed as a creep deflection–time curve, as shown in Figure 9.

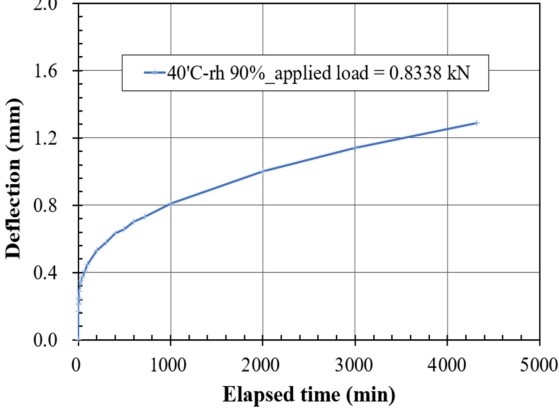

**Figure 9.** Creep deflection–time curve from the box creep test.

## 4. Finite Element Analysis for Multistacked Corrugated Packages

### 4.1. Validation of the FEA Material Properties

To determine the material constants of the core defined in Figure 2, optimization was performed to minimize the errors in the FEA and test using HyperStudy and Abaqus [16,20]. As an FE model, the BCT FE model of the corrugated package and the four-point bending FE models of the MD and CD of the corrugated fiberboard were composed using the Abaqus implicit dynamic quasi-static model.

Similar to the test specifications [17] shown in Figure 5a, the four-point bending FE model is shown in Figure 10. The size of the model was 50 mm × 500 mm, with a thickness of 5 mm, which was the same as those of the specimen used in the test. The element size of the corrugated fiberboard specimen was 5 mm, and it was composed of a continuum shell element to predict the changes over time. The number of nodes and elements in the four-point bending FE model was 2222 and 1000, respectively. In addition, the models of loading and supporting anvils were modeled to be similar to the actual size and shape of the jig (cylinder radius: 3.18 mm; length: 84 mm) during the test and consisted of rigid elements with a size of 2 mm. In the FEA, contact was defined between the corrugated fiberboard specimen and jigs, and the coefficient of friction was determined to be 0.4. For the boundary conditions, the lower supporting anvil was fixed and the upper loading anvil was forcibly displaced in the z-direction. The forced displacement was 15 mm for the CD and 5 mm for the MD, and these values were defined as the average values that could be estimated as linear intervals, regardless of the bending direction of the curves of each specimen's direction, as shown in Figure 5.

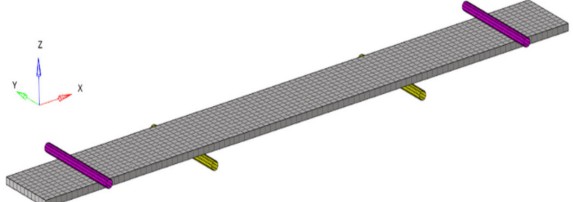

**Figure 10.** Four-point bending FE model of 50 mm × 500 mm.

The BCT FE model is shown in Figure 11. The modeling of the corrugated package was represented in 3D and based on the drawings of the package (external dimension: 730 × 293 × 336 mm), and was applied to the box compression and creep tests. After fabricating the middle surface based on this 3D model, the FE model was constructed in the manner shown in Figure 3. The number of nodes and elements in the BCT FE model was 111,650 and 55,832, respectively. If the degrees of freedom of rotation of the connector element of the box fold are free, convergence becomes difficult during FEA; if the box is not taped, the folded state does not remain because of the reaction force of the fold. Therefore, in this study, nonlinear rotating spring characteristics were applied to the fold to set the initial load due to the spring action in the box assembly (the box flaps were folded at an angle of 90°). In addition, an adhesive tape with a width of 30 mm and a thickness of 0.8 mm was modeled as a membrane element to maintain the folding state of the box, which was composed of rigid elements in the form of a surface at the top and bottom to apply a compressive load on the box. Contact was defined between the box flap and the flap and between the box flaps and the rigid surface, and a friction coefficient of 0.4 was applied [21]. A tie constraint condition was applied between the box and the adhesive tape. As for the condition of the boundary, the lower rigid surface was fixed and the upper rigid surface had a forced displacement of 0.75 mm applied in the z-direction; all other directions were fixed.

HyperStudy was utilized to optimize the material constants [21]. The response set here was the difference in the area of the force–displacement graph of the test and analysis, and the "area_between_two_curves" function was used to calculate this; the "global response search method" was used to minimize this response. The initial value of the material

constants was obtained from the literature [15]. Table 1 lists the material constants of the applied liner.

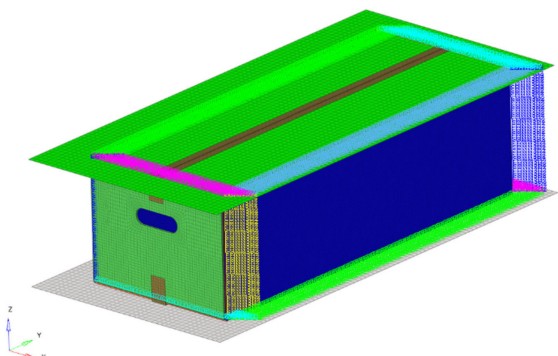

**Figure 11.** Box compression test finite element model of 730 × 293 × 336 mm.

**Table 1.** Material properties of the applied liner.

| $E_1$ | $E_2$ | $v_{12}$ | $G_{12}$ | $G_{13}$ | $G_{23}$ |
|---|---|---|---|---|---|
| 7000 MPa | 3500 MPa | 0.34 | 2000 MPa | 20 MPa | 70 MPa |

Among the material constants of the core, the modulus of elasticity ($E_1$ and $E_2$) and Poisson's ratio ($v_{12}$) were optimized, and the remaining values were obtained from the literature [15]. Table 2 lists the initial material constants of the applied core.

**Table 2.** Initial material properties of the applied core.

| $E_1$ | $E_2$ | $v_{12}$ | $G_{12}$ | $G_{13}$ | $G_{23}$ |
|---|---|---|---|---|---|
| 509.6 MPa | 10 MPa | 0.1978 | 0.231 MPa | 3.5 MPa | 35 MPa |
| Variable | Variable | Variable | Constant | Constant | Constant |

The results of the final area difference after optimization are shown in Table 3, and the comparison results of the reaction force–displacement graphs obtained from the test and FEA are shown in Figure 12a–c.

**Table 3.** Results of the area difference.

| BCT | 4-Point Bending (MD) | 4-Point Bending (CD) |
|---|---|---|
| 3.08 | 6.90 | 0.74 |

CD, count machine direction; MD, machine direction.

Next, optimization was performed for the material constants (*A*, *n*, and *m*) of the creep model using Equation (2). We assumed that the creep behaviors of the liner and core were the same and that the creep FE model of the target packaging box was the same as that of the BCT FE model shown in Figure 11. We combined the FEA and the power-law creep model and simulated static deflection by applying a load of 0.8338 kN in the z-direction in the "implicit dynamic, quasi-static step", and performed creep simulation for 72 h in the "Visco step" [16]. The method of optimization is the same as that performed earlier; the final area difference was 83.8, and the comparison results of the deflection–elapsed time graph are shown in Figure 12d.

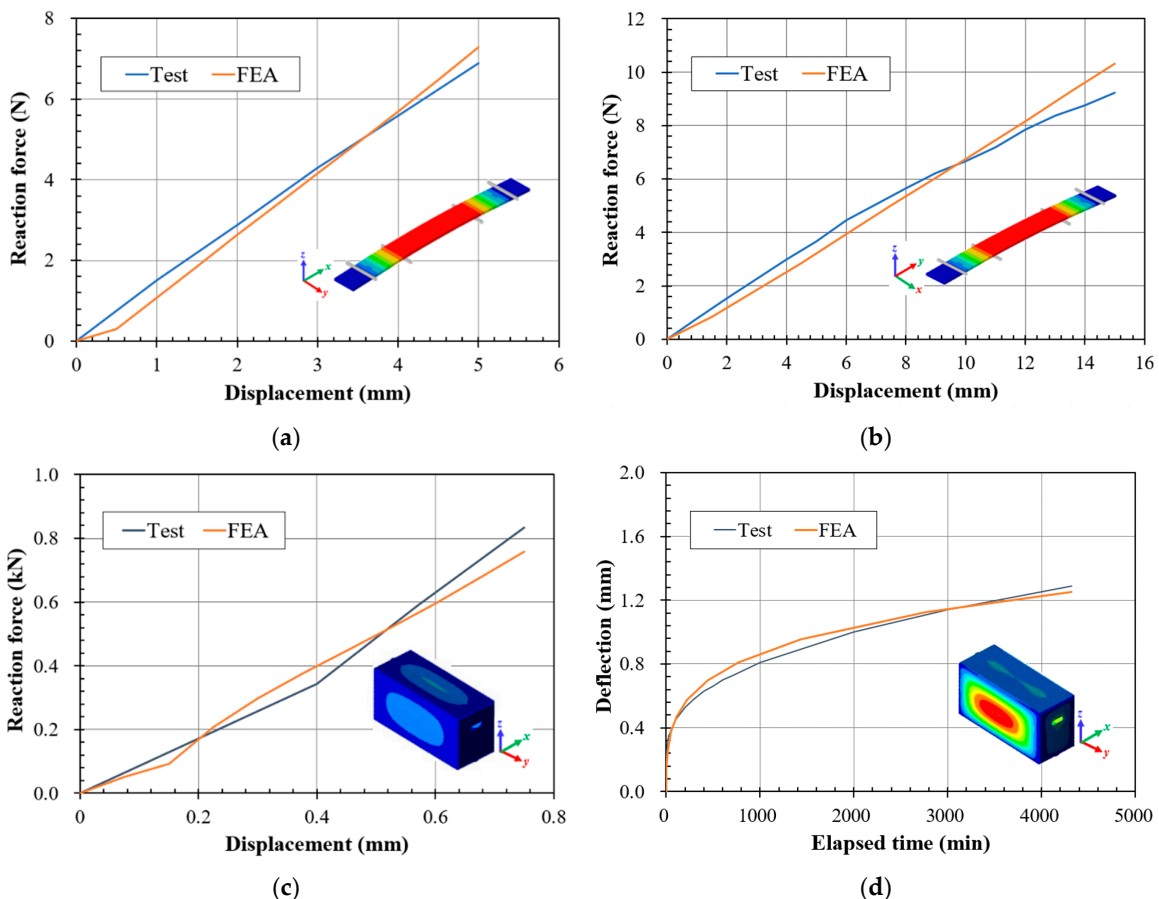

**Figure 12.** Comparison of the FEA and the test results: (**a**) bending_MD; (**b**) bending_CD; (**c**) BCT; and (**d**) box creep test.

### 4.2. FEA for Multistage Loading of the Corrugated Package

Table 4 shows the various scenarios applied during the FEA for the multistage loading of the target corrugated package based on the validation results of the FEA material constants. For FEA, we adopted a method in which a column stacking pattern (the lower part of the stacking) with six packages was fabricated, which was followed by the application of a dummy weight on these packages.

**Table 4.** Scenarios applied in the finite element analysis simulation for the multistacking of a target corrugated package.

| Case | Package Weight (N) | Outside Size (L × W × D) (mm) | Number Stacked | Pallet Weight (N) | Stacking Duration (h) | Stacking Condition |
|------|------|------|------|------|------|------|
| Case 1 | 72 | 730 × 293 × 336 | | | | |
| Case 2 | 78 | 730 × 293 × 336 | 12 | 275 | 72 | 40 °C-rh 90% |
| Case 3 | 98 | 730 × 293 × 233 | | | | |
| Case 4 | 55 | 1083 × 285 × 327 | | | | |

FEA was performed in two stages: after performing a static analysis (implicit dynamic quasi-static analysis) that applied the load to the stacking FE model (for example, in Case 1 of Table 4, the number of nodes = 669,900 (111,650 × 6), and the number of elements = 334,992 (55,830 × 6)) (Figure 13), which consisted of six BCT FE models as shown in Figure 11, the FEA of the creep behavior (Visco) was performed for 72 h. Surface-to-surface contact was defined between the boxes. Park et al. [20] reported that the coefficient of friction between corrugated fiberboards with the same paper combina-

tion and flute shape is not significantly different depending on the contact direction, and the static frictional coefficient between corrugated fiberboards with a paper combination similar to that of the target corrugated fiberboard used in this study was approximately 0.38 (±0.01). Therefore, in this study, the coefficient of friction was determined to be 0.4 without considering the contact direction between the boxes in the FE model, as shown in Figure 13.

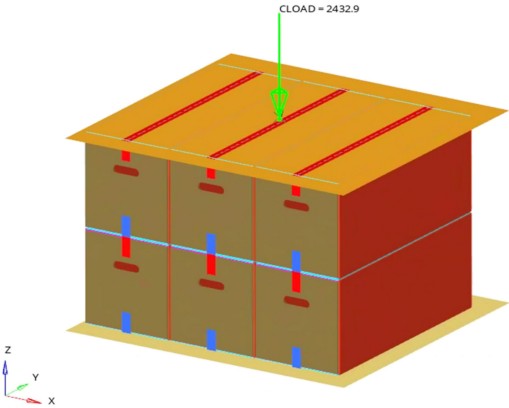

**Figure 13.** Multistacked finite element model of the target corrugated package.

During the test (Figure 14), the height reduction was defined as the average value of the vertical displacement at the top four corners of the weight application. Table 5 shows the FEA results for the various scenarios listed in Table 4, and Figure 15 shows the change in height over time. The applied load in Table 5 was determined by considering the actual distribution conditions of the corrugated package in each case. With the modeling and FEA methods applied in this study, the prediction accuracy of a change in height of the target corrugated package was greater than 80% based on the test value, and the aspect of the change in height over time was similar to the typical creep behavior (elastic stage → transient creep → steady-state creep) of a viscoelastic material.

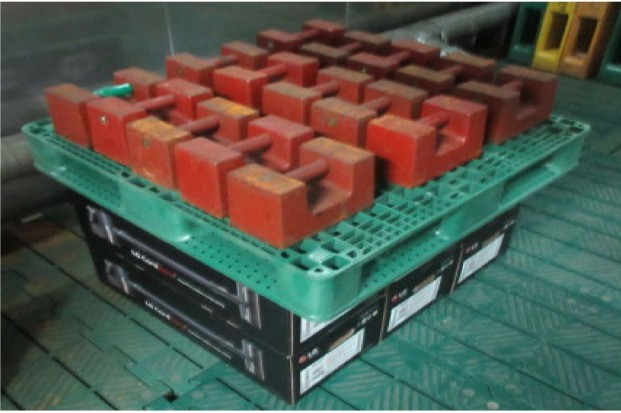

**Figure 14.** Stacking test of the target corrugated package.

**Table 5.** Comparison of the finite element analysis and test results.

| Case | Applied Load (kN) | Outside Size (L × W × D) (mm) | Height Reduction (mm) | |
|---|---|---|---|---|
| | | | Test | FEA |
| Case 1 | 2.4329 | 730 × 293 × 336 | 2.0 (0.61) | 2.1 |
| Case 2 | 2.6291 | 730 × 293 × 336 | 2.8 (0.92) | 2.8 |
| Case 3 | 3.2177 | 730 × 293 × 233 | 5.5 (1.06) | 4.5 |
| Case 4 | 1.9228 | 1083 × 285 × 327 | 0.6 (0.19) | 0.4 |

() standard deviation.

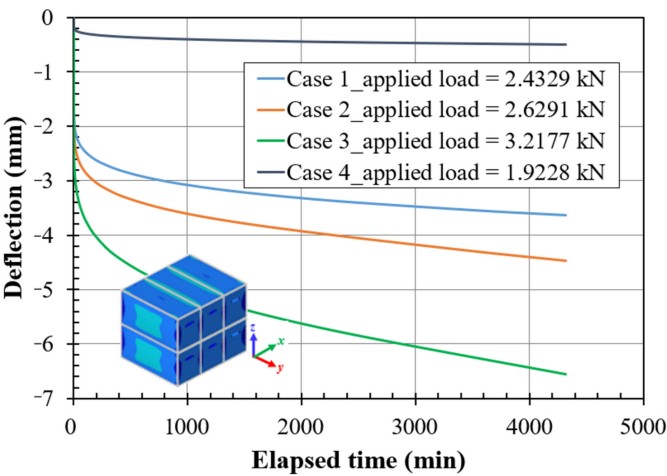

**Figure 15.** Finite element analysis results of a multistacked target corrugated package.

## 5. Conclusions

CAE prediction technology for determining compression behavior in multistacked corrugated packages is an important tool for preventing warehouse accidents, shortening packaging development periods, and optimizing packaging. In this study, to predict the height change over time of the multistacked target corrugated package, the existing package simulation method was investigated, supplemented, and then applied to this study.

All simulations were performed with Abaqus. The power-law creep model was applied to describe the height change over time of the corrugated package, and it was composed of a composite model to show the laminated anisotropy characteristics of the target corrugated fiberboard (double wall of EB-flute). Basically, the continuum shell element type that is advantageous for a change over time and contact was applied, and shell-to-solid coupling and connector elements were used for modeling to describe the proper behavior of the box fold. To determine the material constant required for the FEA simulation, four-point bending, box compression, and box creep tests were performed. A simulation model that simulates the test equally was constructed, and parameter optimization that minimized the area difference was performed using HyperStudy through an area comparison of the F–D curve of the test and the FEA. Based on parameter optimization, the FEA simulation and stacking test for the multistacked target corrugated package were performed simultaneously on four actual stacking scenarios with different package weights and package sizes. A comparison of height changes after 72 h of stacking for each of the four scenarios showed that the concordance between the test and FEA simulation was more than 80% in all cases.

It is planned that, in the future, the FEA methodology described in this study will be used in the development of a virtual verification process for corrugated packages. To further expand the scope of this application, it is necessary to secure additional reliability through continuous comparative monitoring using the test data and physical properties of various corrugated fiberboards.

**Author Contributions:** Data curation, J.-M.P. and H.-M.J.; formal analysis, J.-M.P. and T.-Y.P.; funding acquisition, H.-M.J.; investigation, H.-M.J.; software, J.-M.P. and T.-Y.P.; validation, J.-M.P. and T.-Y.P.; visualization, J.-M.P. and T.-Y.P.; writing—original draft, J.-M.P. and H.-M.J.; writing—review and editing, J.-M.P. and H.-M.J.; supervision, H.-M.J. All authors have read and agreed to the published version of the manuscript.

**Funding:** This research was carried out with the support of the Cooperative Research Program for the development of data application technology for the postharvest management of agricultural and livestock products (Project No.: PJ017050012023); Rural Development Administration.

**Institutional Review Board Statement:** Not applicable.

**Informed Consent Statement:** Not applicable.

**Conflicts of Interest:** The authors declare no conflict of interest.

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
