# Peer review of "Prediction of Deflection Due to Multistage Loading of a Corrugated Package"

_applsci, doi:10.3390/app13074236_

Round 1
Reviewer 1 Report
Comments for the paper titled by “Prediction of Deflection due to Multistage Loading of a Corrugated Package”
In the manuscript, the authors used material property tests to achieve the material constants, which applied to establish the Finite Element Model of a corrugated package. The comparation between FEA simulation and stacking test was within the margin of error and the change behavior in height over time was obtained. However, there are still some issues need to be addressed before considering possible publication. The major comments and suggestions are listed as follows.
1. Figure 12 in page 10, the simulation results in (c) and (d) clearly show that the maximum damage to the corrugated packaging is at the center of the side, so why is the average depth of downward pressure at the four vertices on the upper surface used as the response indicator instead of the depth of depression at the center of the side?
2. The results about the prediction of deflection need be talking deeply, why the third group of tests in Table 5 has a large error with the simulation, and whether a larger load would be an inaccurate prediction.
3. The fourth group of tests in Table 5 should give detailed values rather than a range.
4. Some sentences in the article are not clearly expressed, which need to be revised. Therefore, it is recommended that the entire text be checked in English.
5. Line 80 and line 83 in page 2, Formulas should be centered using tabs.
6. Whole manuscript line graphs should have the grid removed to ensure readability, for example, the bending moment sketch in the upper left corner of Figure 5(a) is affected by the grid.
7. The labels (a), (b), (c), and (d) in Figure 12 should be centered below their respective line graphs.
8. Line 154 in page 5, the space between “40” and “℃” should be deleted, please check and revise the whole manuscript.
9. Please keep the figure format consistent throughout the manuscript.
Reviewer 2 Report
The manuscript is well written. Results are clear.
The authors should provide the number of elements they used in all the simulations.
More details should be given regarding the shell-to solid coupling.
In table 2 correct the misspelled word 'vaiable'.
Reviewer 3 Report
1- The abstract should be written in clear and understandable sentences.
2- What is done in this study, which is different from the literature, should be clearly explained.
3- ''In this study, the DW corrugated fiberboard of the EB-flute (EB/F) was targeted.'' The part that starts with this sentence is very difficult to understand. In addition, if there is a source where the geometric properties of the cardboard types expressed here are given before, it should be stated.
4- E/F, F/E.... what do they represent? If it is related to flutes, this should be explained to the reader.
5- If a side view of the produced sample is given, the flute parts can be seen.
6-Details should be given about the material produced.
7-Why is the FEM drawing shown in Figure 4 given?
8- The sample used in Figure 6 should also be shown.
9- I couldn't interpret where it says height reduction less than 1 while the outer dimension is 1083 in Table 5. Taken again, it may be understandable to all readers.
The 10-Results section should be reinterpreted. The results of the study should be clearly explained.
11-The part expressed with more than 80% accuracy should be detailed in the text.
Round 2
Reviewer 1 Report
Responses over reviewer's comments are well addressed. The paper is worth to be published.
Reviewer 3 Report
The results section and the abstract should be reinterpreted. The results of the study should be clearly explained.
It is not said that there is a typo in English in this part. It was asked to be rewritten only to make it more understandable again.
